# Warm Start Marginal Likelihood Optimisation for Iterative Gaussian Processes

**Jihao Andreas Lin**[1,2]       JAL232@CAM.AC.UK

**Shreyas Padhy**[1]       SP2058@CAM.AC.UK

**Bruno Mlodozeniec**[1,2]       BKM28@CAM.AC.UK

**José Miguel Hernández-Lobato**[1]       JMH233@CAM.AC.UK

[1]*University of Cambridge*      [2]*Max Planck Institute for Intelligent Systems*

## 1. Introduction

Gaussian processes (Rasmussen and Williams, 2006) are a versatile probabilistic machine learning model that have found great success in many applications, such as Bayesian optimisation of black-box functions (Snoek et al., 2012) or data-efficient learning in robotics and control (Deisenroth et al., 2015). However, their effectiveness often depends on performing model selection, which amounts to finding good estimates of quantities such as kernel hyperparameters, and the amount of observation noise prescribed in the likelihood. For Gaussian processes, these quantities are typically learned by maximising the marginal likelihood of the training data, which balances the expressiveness and the complexity of a model in representing the training data. Unfortunately, conventional approaches which use the Cholesky factorisation have limited scalability, because the computational costs and memory requirements are respectively cubic and quadratic in the amount of training data.

Many methods have been developed to improve the scalability of Gaussian processes. Typically, they either leverage a handful of judiciously chosen inducing points to represent the training data sparsely; or solve large systems of linear equations using iterative methods. Sparse methods (Quiñonero-Candela and Rasmussen, 2005; Titsias, 2009; Hensman et al., 2013) are fundamentally limited in the number of inducing points, because the same cubic and quadratic scaling of compute and memory requirements still applies to the number of inducing points. With increasingly large or sufficiently complex data, a limited number of inducing points can no longer accurately represent the original data. In contrast, iterative methods (Gardner et al., 2018; Lin et al., 2023; Wu et al., 2024) attempt to solve the original problem up to a specified numerical precision, therefore allowing a trade-off between compute time and accuracy of a solution. Nonetheless, they can be slow in the large data regime due to slow convergence properties, sometimes requiring several days of training time despite leveraging parallel compute capabilities (Wang et al., 2019).

In this work, we consider marginal likelihood optimisation for iterative Gaussian processes. We introduce a three-level hierarchy of marginal likelihood optimisation for iterative Gaussian processes (Figure 2), and identify that the computational costs are dominated by solving sequential batches of large positive-definite systems of linear equations (Figure 3). We then propose to amortise computations by reusing solutions of linear system solvers as initialisations in the next step, providing a *warm start*. Finally, we discuss the necessary conditions and quantify the consequences of warm starts (Theorem 1) and demonstrate their effectiveness on regression tasks (Table 1), where warm starts achieve the same results as the conventional procedure while providing up to a $16\times$ average speed-up among datasets.

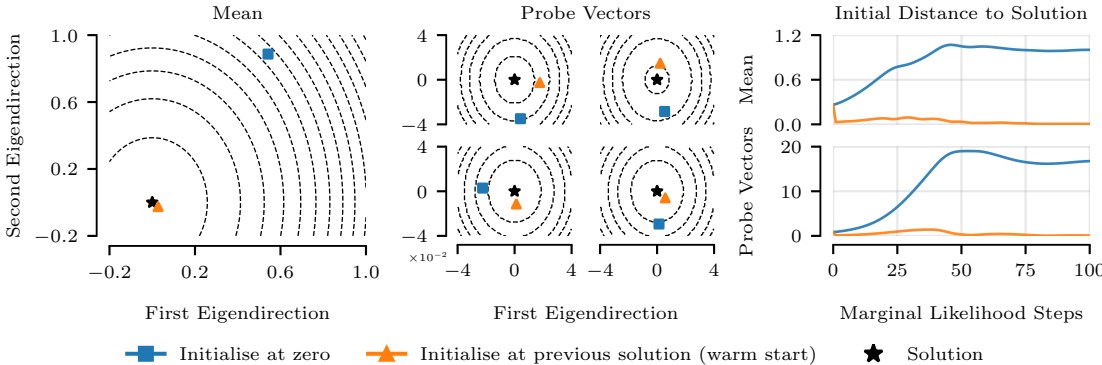

Figure 1: Two-dimensional cross-sections of quadratic objectives targeted by linear solvers after twenty marginal likelihood steps on the POL dataset, centred at the solution and visualised along eigendirections corresponding to the two largest eigenvalues (left), and evolution of the distance between initialisation and solution measured as root-mean-square error with respect to the norm induced by the curvature of the quadratic objective (right). Initialising at the previous solution (warm start) substantially reduces the initial distance to the solution.

## 2. Gaussian Process Regression and Marginal Likelihood Optimisation

Formally, a Gaussian process is a stochastic process $f : \mathcal{X} \to \mathbb{R}$, such that, for any finite subset $\{x_i\}_{i=1}^n \subset \mathcal{X}$, the set of random variables $\{f(x_i)\}_{i=1}^n$ follows a multivariate Gaussian distribution. In particular, $f$ is uniquely identified by a mean function $\mu(\cdot) = \mathbb{E}[f(\cdot)]$ and a positive-definite kernel function $k(\cdot, \cdot'; \vartheta) = \mathrm{Cov}(f(\cdot), f(\cdot'))$ with kernel hyperparameters $\vartheta$. We write $f \sim \mathrm{GP}(\mu, k)$ to express that $f$ is a Gaussian process with mean $\mu$ and kernel $k$.

For the purpose of Gaussian process regression, let the training data consist of $n$ inputs $\boldsymbol{x} \subset \mathcal{X}$ and corresponding targets $\boldsymbol{y} \subset \mathbb{R}$. We consider the Bayesian model $y_i = f(x_i) + \epsilon_i$, where each $\epsilon_i \sim \mathcal{N}(0, \sigma^2)$ identically and independently, and $f \sim \mathrm{GP}(\mu, k)$, where we assume $\mu = 0$ without loss of generality. The posterior of this model is $f|\boldsymbol{y} \sim \mathrm{GP}(\mu_{f|\boldsymbol{y}}, k_{f|\boldsymbol{y}})$, with

$$\mu_{f|\boldsymbol{y}}(\cdot) = k(\cdot, \boldsymbol{x}; \vartheta)(k(\boldsymbol{x}, \boldsymbol{x}; \vartheta) + \sigma^2 \mathbf{I})^{-1} \boldsymbol{y}, \tag{1}$$

$$k_{f|\boldsymbol{y}}(\cdot, \cdot') = k(\cdot, \cdot'; \vartheta) - k(\cdot, \boldsymbol{x}; \vartheta)(k(\boldsymbol{x}, \boldsymbol{x}; \vartheta) + \sigma^2 \mathbf{I})^{-1} k(\boldsymbol{x}, \cdot'; \vartheta), \tag{2}$$

where $k(\cdot, \boldsymbol{x}; \vartheta)$, $k(\boldsymbol{x}, \cdot; \vartheta)$ and $k(\boldsymbol{x}, \boldsymbol{x}; \vartheta)$ refer to pairwise evaluations, resulting in a $1 \times n$ row vector, a $n \times 1$ column vector and a $n \times n$ matrix respectively.

With $\boldsymbol{\theta} = \{\vartheta, \sigma\}$ and $\mathbf{H}_{\boldsymbol{\theta}} = k(\boldsymbol{x}, \boldsymbol{x}; \vartheta) + \sigma^2 \mathbf{I}$, the marginal likelihood $\mathcal{L}$ as a function of $\boldsymbol{\theta}$ and its gradient $\nabla \mathcal{L}$ with respect to $\boldsymbol{\theta}$ can be expressed as

$$\mathcal{L}(\boldsymbol{\theta}) = -\frac{1}{2} \boldsymbol{y}^{\mathsf{T}} \mathbf{H}_{\boldsymbol{\theta}}^{-1} \boldsymbol{y} - \frac{1}{2} \log \det \mathbf{H}_{\boldsymbol{\theta}} - \frac{n}{2} \log 2\pi, \tag{3}$$

$$\nabla \mathcal{L}(\boldsymbol{\theta}) = \frac{1}{2} (\mathbf{H}_{\boldsymbol{\theta}}^{-1} \boldsymbol{y})^{\mathsf{T}} \frac{\partial \mathbf{H}_{\boldsymbol{\theta}}}{\partial \boldsymbol{\theta}} \mathbf{H}_{\boldsymbol{\theta}}^{-1} \boldsymbol{y} - \frac{1}{2} \mathrm{tr} \left( \mathbf{H}_{\boldsymbol{\theta}}^{-1} \frac{\partial \mathbf{H}_{\boldsymbol{\theta}}}{\partial \boldsymbol{\theta}} \right), \tag{4}$$

where the partial derivative of $\mathbf{H}_{\boldsymbol{\theta}}$ with respect to each element in $\boldsymbol{\theta}$ is a $n \times n$ matrix. If $n$ is small enough such that a Cholesky factorisation of $\mathbf{H}_{\boldsymbol{\theta}}$ is tractable then both $\mathcal{L}$ and $\nabla \mathcal{L}$ can be easily evaluated and used by any optimiser of choice to maximise $\mathcal{L}$. However, we are considering the case where $n$ is too large to compute the Cholesky factorisation of $\mathbf{H}_{\boldsymbol{\theta}}$.

### 2.1. Marginal Likelihood Optimisation for Iterative Gaussian Processes

Marginal likelihood optimisation in iterative Gaussian processes can be structured into a three-level hierarchy (see Figure 2), as follows.

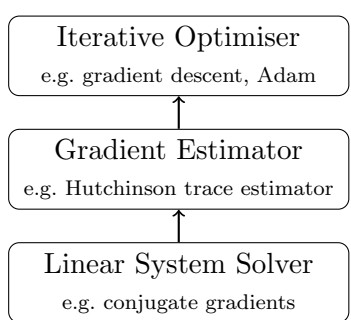

**Iterative Optimiser**  Typically, a first-order optimiser, such as Adam (Kingma and Ba, 2015), is used to maximise $\mathcal{L}$, which only requires estimates of $\nabla\mathcal{L}$, avoiding the evaluation of $\mathcal{L}$ and $\log\det\mathbf{H_\theta}$. This allows us to focus on tractable estimates of $\nabla\mathcal{L}$.

Figure 2: Marginal likelihood optimisation framework for iterative Gaussian processes.

**Gradient Estimator**  The gradient $\nabla\mathcal{L}$ (4) involves two computationally expensive components: inverse matrix-vector products of the form $\boldsymbol{v} = \mathbf{H_\theta}^{-1}\boldsymbol{b}$ and the trace term. The inverse matrix-vector products are readily approximated using iterative solvers to linear systems of the form $\mathbf{H_\theta}\boldsymbol{v} = \boldsymbol{b}$. The trace term can also be reduced into inverse matrix-vector products using stochastic trace estimation, e.g. Hutchinson's (Hutchinson, 1990), as follows

$$\mathrm{tr}\left(\mathbf{H_\theta}^{-1}\frac{\partial\mathbf{H_\theta}}{\partial\boldsymbol{\theta}}\right) = \mathbb{E}_{\boldsymbol{z}}\left[\boldsymbol{z}^\mathsf{T}\mathbf{H_\theta}^{-1}\frac{\partial\mathbf{H_\theta}}{\partial\boldsymbol{\theta}}\boldsymbol{z}\right] \approx \frac{1}{s}\sum_{j=1}^{s}\boldsymbol{z}_j^\mathsf{T}\mathbf{H_\theta}^{-1}\frac{\partial\mathbf{H_\theta}}{\partial\boldsymbol{\theta}}\boldsymbol{z}_j, \tag{5}$$

where $s$ probe vectors $\boldsymbol{z}_j$ of length $n$ are introduced and $\forall j : \mathbb{E}[\boldsymbol{z}_j\boldsymbol{z}_j^\mathsf{T}] = \mathbf{I}$ is required for the estimator to be unbiased. Common choices for the distribution of $\boldsymbol{z}_j$ are standard Gaussian, $\boldsymbol{z}_j \sim \mathcal{N}(\mathbf{0}, \mathbf{I})$, or Rademacher, namely uniform random signs, $\boldsymbol{z}_j \sim \mathcal{U}(\{1, -1\})^n$. In theory, the latter exhibit lower estimator variance. Additionally, more advanced trace estimators have also been developed (Meyer et al., 2021; Epperly et al., 2024). However, in practice, standard Gaussian probes with Hutchinson's trace estimator seem to work well.

**Linear System Solver**  After substituting the trace estimator into (4), the approximate gradient consists of terms that involve computing $\boldsymbol{v_y} = \mathbf{H_\theta}^{-1}\boldsymbol{y}$ and $\boldsymbol{v}_j = \mathbf{H_\theta}^{-1}\boldsymbol{z}_j$ by solving large systems of linear equations,

$$\mathbf{H_\theta}\left[\boldsymbol{v_y}, \boldsymbol{v}_1, \ldots, \boldsymbol{v}_s\right] = \left[\boldsymbol{y}, \boldsymbol{z}_1, \ldots, \boldsymbol{z}_s\right], \tag{6}$$

which share the same coefficient matrix $\mathbf{H_\theta}$. Because $\mathbf{H_\theta}$ is positive-definite, the solution $\boldsymbol{v} = \mathbf{H_\theta}^{-1}\boldsymbol{b}$ to the system of linear equations $\mathbf{H_\theta}\boldsymbol{v} = \boldsymbol{b}$ can also be obtained by finding the unique minimiser of the corresponding convex quadratic objective,

$$\boldsymbol{v} = \arg\min_{\boldsymbol{u}} \ \frac{1}{2}\boldsymbol{u}^\mathsf{T}\mathbf{H_\theta}\,\boldsymbol{u} - \boldsymbol{u}^\mathsf{T}\boldsymbol{b}, \tag{7}$$

facilitating the use of iterative optimisers. In the context of Gaussian processes, conjugate gradients (Gardner et al., 2018; Wang et al., 2019; Wilson et al., 2020, 2021), alternating projections (Wu et al., 2024) and stochastic gradient descent (Lin et al., 2023, 2024) have been applied to optimise (7), serving as linear system solvers.

Notably, the linear system solver dominates the overall computational costs, such that reducing its runtime translates to substantial computational savings (see Figure 3). Therefore, we propose to amortise computations by reusing solutions of linear systems to initialise the linear system solver in the next marginal likelihood step, providing a *warm start*.

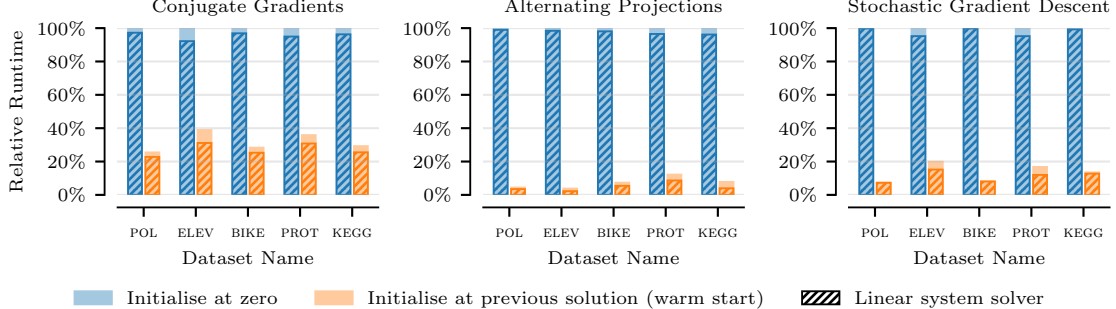

Figure 3: Comparison of relative runtimes for different linear system solvers. The solver (striped areas) dominates the total training time (coloured patches). Initialising at the previous solution (warm start) significantly reduces the runtime of the linear system solver, with varying effectiveness among different solvers and datasets.

## 3. Warm Start Marginal Likelihood Optimisation

Given that the iterative linear system solver dominates the computational costs of marginal likelihood optimisation (see Figure 3), reducing the number of necessary solver iterations until convergence will translate to substantial computational savings. However, iterative solvers are typically initialised at zero for each gradient computation step, even though the hyperparameters do not change much between steps.[1] Therefore, we propose to amortise computational costs for any solver type by reusing solutions of previous linear systems to *warm start* (i.e. initialise) linear system solvers in the subsequent step.

At iterations $t$ and $t+1$ of the marginal likelihood optimiser, associated with $\boldsymbol{\theta}^{(t)}$ and $\boldsymbol{\theta}^{(t+1)}$, the linear system solver must solve two batches of linear systems, namely

$$\mathbf{H}_{\boldsymbol{\theta}}^{(t)} \left[ \boldsymbol{v}_{\boldsymbol{y}}^{(t)}, \boldsymbol{v}_1^{(t)}, \ldots, \boldsymbol{v}_s^{(t)} \right] = \left[ \boldsymbol{y}, \boldsymbol{z}_1^{(t)}, \ldots, \boldsymbol{z}_s^{(t)} \right] \quad \text{and} \tag{8}$$

$$\mathbf{H}_{\boldsymbol{\theta}}^{(t+1)} \left[ \boldsymbol{v}_{\boldsymbol{y}}^{(t+1)}, \boldsymbol{v}_1^{(t+1)}, \ldots, \boldsymbol{v}_s^{(t+1)} \right] = \left[ \boldsymbol{y}, \boldsymbol{z}_1^{(t+1)}, \ldots, \boldsymbol{z}_s^{(t+1)} \right], \tag{9}$$

where $\mathbf{H}_{\boldsymbol{\theta}}^{(t)}$ and $\mathbf{H}_{\boldsymbol{\theta}}^{(t+1)}$ are related through the change from $\boldsymbol{\theta}^{(t)}$ to $\boldsymbol{\theta}^{(t+1)}$ and $\boldsymbol{v}_{\boldsymbol{y}}^{(t)}$ and $\boldsymbol{v}_{\boldsymbol{y}}^{(t+1)}$ are further related through sharing the same right-hand side $\boldsymbol{y}$ in the linear system. In such a setting, where the coefficient matrix only changes slightly and the right-hand side remains fixed, we can approximate $\boldsymbol{v}^{(t+1)}$ using a first-order Taylor expansion of $\mathbf{H}_{\boldsymbol{\theta}}^{(t+1)}$,

$$\left( \mathbf{H}_{\boldsymbol{\theta}}^{(t+1)} \right)^{-1} \approx \left( \mathbf{H}_{\boldsymbol{\theta}}^{(t)} \right)^{-1} - \left( \mathbf{H}_{\boldsymbol{\theta}}^{(t)} \right)^{-1} \left( \mathbf{H}_{\boldsymbol{\theta}}^{(t+1)} - \mathbf{H}_{\boldsymbol{\theta}}^{(t)} \right) \left( \mathbf{H}_{\boldsymbol{\theta}}^{(t)} \right)^{-1}, \tag{10}$$

$$\boldsymbol{v}^{(t+1)} \approx \boldsymbol{v}^{(t)} - \left( \mathbf{H}_{\boldsymbol{\theta}}^{(t)} \right)^{-1} \left( \mathbf{H}_{\boldsymbol{\theta}}^{(t+1)} - \mathbf{H}_{\boldsymbol{\theta}}^{(t)} \right) \boldsymbol{v}^{(t)}. \tag{11}$$

If $\Delta = \mathbf{H}_{\boldsymbol{\theta}}^{(t+1)} - \mathbf{H}_{\boldsymbol{\theta}}^{(t)}$ is small then $\boldsymbol{v}^{(t)}$ will be close to $\boldsymbol{v}^{(t+1)}$ (see Figure 1), such that we can reuse $\boldsymbol{v}^{(t)}$ to initialise the linear system solver when solving for $\boldsymbol{v}^{(t+1)}$. To satisfy the condition of fixed right-hand sides, we propose to set $\boldsymbol{z}_j^{(t)} = \boldsymbol{z}_j$ at the cost of introducing some bias throughout optimisation, which can be bounded, as we will now quantify.

---

1. Notable exceptions are Artemev et al. (2021), who warm start $\boldsymbol{v}_{\boldsymbol{y}}$ in a sparse lower bound on $\mathcal{L}$, and Antorán et al. (2023), who warm start a stochastic gradient descent solver for generalised linear models.

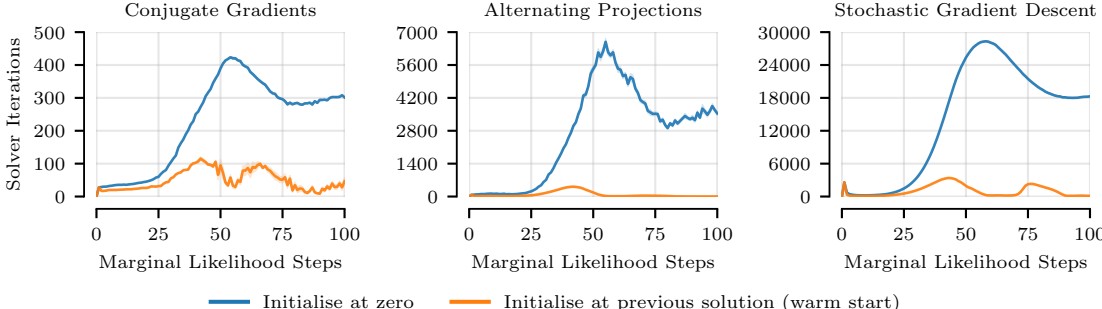

Figure 4: Evolution of the required number of linear system solver iterations at each step of marginal likelihood optimisation on the POL dataset. Initialising at the solution of the previous step (warm start) reduces the number of required solver iterations with varying effectiveness among different solvers.

**Theorem 1** *Let $\mathcal{L}$ and $\nabla\mathcal{L}$ be the marginal likelihood and its gradient as defined in* (3) *and* (4) *respectively, and let $\tilde{g}$ be an approximation to the gradient $\nabla\mathcal{L}$ where the trace is approximated with s fixed samples as in* (5)*. Assume that the hyperparameter optimisation domain $\Theta$ is convex, closed and bounded, and that $\tilde{g} : \Theta \to \mathbb{R}$ is a conservative field. Then, given a sufficiently large number of samples s, the hyperparameters $\tilde{\theta}^*$ obtained by maximising the objective implied by the approximate gradients $\tilde{g}$ will be $\gamma$-close in terms of the true objective $\mathcal{L}$ to the true maximum $\theta^*$ of the objective $\mathcal{L}$,*

$$\mathcal{L}(\tilde{\theta}^*) \geq \mathcal{L}(\theta^*) - \gamma,$$

*with probability at least $1 - \delta$.*

See Appendix A for details. In practice, a small number of samples seems to be sufficient.

## 4. Experiments

To investigate the effectiveness of warm starts, we performed marginal likelihood optimisation on five UCI regression datasets (Dua and Graff, 2017), comparing the procedure described in Section 2.1 with resampled probe vectors versus fixed probe vectors and warm starts. In particular, we used the Matérn-$3/2$ kernel with length scales per input dimension and a scalar signal scale. Observation noise, signal scale and length scales were initialised at 1.0 and jointly optimised by performing 100 steps of Adam (Kingma and Ba, 2015) with a learning rate of 0.1, where the gradient was estimated using (5) with $s = 16$ standard Gaussian probe vectors $z_j \sim \mathcal{N}(0, I)$. We conducted experiments with different linear system solvers, namely conjugate gradients (Gardner et al., 2018; Wang et al., 2019), alternating projections (Wu et al., 2024) and stochastic gradient descent (Lin et al., 2023, 2024). See Appendix B for further implementation details.

Figure 3 illustrates that warm starts significantly reduce the runtime of linear system solvers, and consequently the total runtime, being most effective for alternating projections. Figure 4 visualises that these speed-ups are due to substantial decreases in the number of linear system solver iterations required to reach a specified tolerance. Table 1 reports the final test log-likelihoods, computed using Cholesky factorisation, and total runtimes after 100 steps of marginal likelihood optimisation. Warm starts achieve the same test performance

Table 1: Predictive test log-likelihoods and total runtimes after marginal likelihood optimisation, and average speed-up among datasets due to warm start for different linear system solvers, namely conjugate gradients (CG), alternating projections (AP), and stochastic gradient descent (SGD) (mean over 10 dataset splits).

| | Test Log-Likelihood | | | | | Total Runtime (min) | | | | | Average |
| | POL | ELEV | BIKE | PROT | KEGG | POL | ELEV | BIKE | PROT | KEGG | Speed-Up |
|---|---|---|---|---|---|---|---|---|---|---|---|
| CG | 1.27 | -0.39 | 2.15 | -0.59 | 1.08 | 7.86 | 2.76 | 7.69 | 31.44 | 64.29 | — |
| + ws | 1.27 | -0.39 | 2.15 | -0.59 | 1.08 | 2.00 | 1.07 | 2.18 | 11.27 | 18.81 | 3.2 × |
| AP | 1.27 | -0.39 | 2.15 | -0.59 | 1.08 | 22.39 | 13.55 | 12.31 | 45.42 | 62.24 | — |
| + ws | 1.27 | -0.39 | 2.15 | -0.59 | 1.08 | 0.99 | 0.52 | 0.90 | 5.52 | 4.86 | 16.7 × |
| SGD | 1.27 | -0.39 | 2.18 | -0.59 | 1.08 | 41.31 | 4.92 | 81.84 | 46.92 | 360.44 | — |
| + ws | 1.27 | -0.39 | 2.15 | -0.59 | 1.07 | 3.08 | 0.98 | 6.73 | 7.87 | 48.72 | 8.8 × |

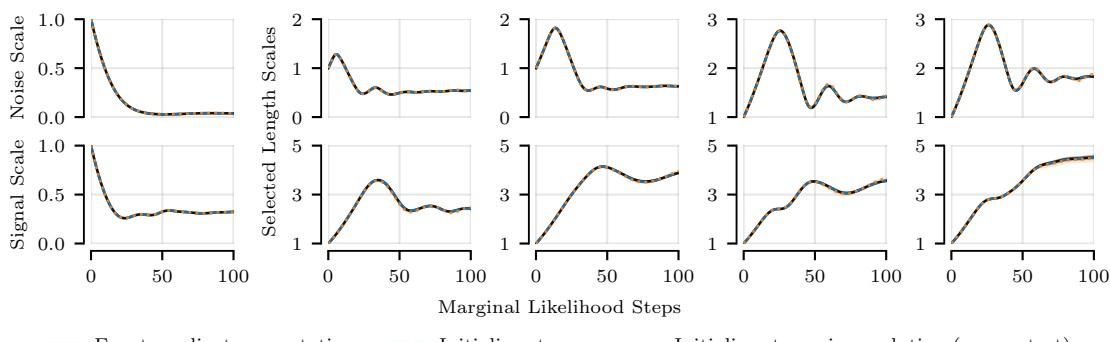

Figure 5: Evolution of hyperparameters during marginal likelihood optimisation on the POL dataset using conjugate gradients as linear system solver. The behaviour of exact gradient computation using Cholesky factorisation is obtained when initialising at zero or at the previous solution. The latter does not degrade performance.

while providing an average speed-ups among datasets from 3.2× to 16.7×, showing that fixing probe vectors and reusing solutions does not impact performance in practice. Figure 5 shows that optimisation traces obtained using warm starts are almost identical to the traces obtained by resampling probe vectors and reinitialising at zero, and exact gradient computation using Cholesky factorisation. See Appendix C for more experimental results.

## 5. Conclusion

We discussed marginal likelihood optimisation for iterative Gaussian processes and proposed warm starts to amortise linear system solver computation. We analysed the consequences of warm starts theoretically, and investigated their behaviour during hyperparameter optimisation on regression tasks empirically. Our experiments demonstrated that warm starts provide substantial reductions in computational costs, while maintaining predictive performance and matching optimisation traces of exact gradient computation using Cholesky factorisation.

## Acknowledgments

Jihao Andreas Lin and Shreyas Padhy were supported by the University of Cambridge Harding Distinguished Postgraduate Scholars Programme. José Miguel Hernández-Lobato acknowledges support from a Turing AI Fellowship under grant EP/V023756/1. We thank Javier Antorán and Runa Eschenhagen for helpful discussions. This work was performed using resources provided by the Cambridge Service for Data Driven Discovery (CSD3) operated by the University of Cambridge Research Computing Service (www.csd3.cam.ac.uk), provided by Dell EMC and Intel using Tier-2 funding from the Engineering and Physical Sciences Research Council (capital grant EP/T022159/1), and DiRAC funding from the Science and Technology Facilities Council (www.dirac.ac.uk).

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

# Appendix A. Mathematical Derivations

Throughout this appendix, we will denote the number of data examples as $n$, such that $\mathbf{H}_{\boldsymbol{\theta}} \in \mathbb{R}^{n \times n}$, and the number of samples in the trace estimator in (5) with $s$. We will denote the optimisation domain for the hyperparameters as $\Theta$, where we assume $\Theta \subseteq \mathbb{R}^{d_\theta}$. We will also assume that all elements in $\boldsymbol{z}_j$ have finite fourth moments $\mathbb{E} \, z^4$. For standard normal $\boldsymbol{z}_j$, we have $\mathbb{E} \, z^4 = 3$, and for Rademacher $\boldsymbol{z}_j$ we have $\mathbb{E} \, z^4 = 1$. Furthermore, we assume that the coordinates of $\boldsymbol{z}_j$ are also pairwise independent, which will again be the case for Gaussian or Rademacher random variables.

**Theorem 2** *Let $\boldsymbol{g} = \nabla \mathcal{L}$, as in (4), and let $\tilde{\boldsymbol{g}} : \Theta \to \mathbb{R}^{d_\theta}$ be an approximation to $\boldsymbol{g}$ with $s$ samples, as in (5). Assume that the absolute value of the eigenvalues of $\mathbf{H}_{\boldsymbol{\theta}}^{-1}, \frac{\partial \mathbf{H}_{\boldsymbol{\theta}}}{\partial \theta_k}$ are upper-bounded on the domain of $\theta_k$ by $\lambda_{\mathbf{H}^{-1}}^{\max}$ and $\lambda_{\partial \mathbf{H}}^{\max}$ such that the eigenvalues of their product are upper-bounded by $\lambda^{\max} = \lambda_{\mathbf{H}^{-1}}^{\max} \lambda_{\partial \mathbf{H}}^{\max}$. Then, for any $\beta, \delta > 0$, if the number of samples is sufficiently large, we have*

$$\mathbb{P}\left[ \left| \tilde{g}_k(\theta) - \frac{\partial}{\partial \theta_k} \mathcal{L}(\theta) \right| > \beta \right] < \delta \qquad if \qquad s > \left(1 + \frac{2}{\epsilon}\right)^n \frac{\mathbb{E}\left[z^4\right] + n - 2}{\delta \beta^2 (1-\epsilon)^2} n \lambda^{\max}, \qquad (12)$$

*i.e. the $j$-th component of the approximate gradient $\tilde{\boldsymbol{g}}(\theta)$ will be within distance $\beta$ of the true gradient on the entire optimisation space $\Theta$ with probability at least $(1 - \delta)$ for any $\epsilon > 0$.*

**Proof** Let $\sum_{i=1}^n \boldsymbol{q}_i(\boldsymbol{\theta}) \lambda_i(\boldsymbol{\theta}) \boldsymbol{p}_i(\boldsymbol{\theta})^\mathsf{T}$ be the eigendecomposition of $\mathbf{H}_{\boldsymbol{\theta}}^{-1} \frac{\partial \mathbf{H}_{\boldsymbol{\theta}}}{\partial \theta_k}$, where $\{\boldsymbol{q}_i\}_{i=1}^n$ and $\{\boldsymbol{p}_i\}_{i=1}^n$ are two sets of orthonormal vectors. We will notationally suppress the dependence of $\boldsymbol{p}_i, \boldsymbol{q}_i, \lambda_i$ on $\boldsymbol{\theta}$ going forwards. First, we rewrite $\tilde{g}_k(\theta) - g_k(\theta)$,

$$\tilde{g}_k(\theta) - g_k(\theta) = \sum_{j=1}^s \boldsymbol{z}_j^\mathsf{T} \mathbf{H}_{\boldsymbol{\theta}}^{-1} \frac{\partial \mathbf{H}_{\boldsymbol{\theta}}}{\partial \theta_k} \boldsymbol{z}_j - \mathbb{E}_{\boldsymbol{z}} \left[ \boldsymbol{z}^\mathsf{T} \mathbf{H}_{\boldsymbol{\theta}}^{-1} \frac{\partial \mathbf{H}_{\boldsymbol{\theta}}}{\partial \theta_k} \boldsymbol{z} \right], \qquad (13)$$

$$= \sum_{j=1}^s \boldsymbol{z}_j^\mathsf{T} \left( \sum_{i=1}^n \lambda_i \boldsymbol{q}_i \boldsymbol{p}_i^\mathsf{T} \right) \boldsymbol{z}_j - \mathbb{E}_{\boldsymbol{z}} \left[ \boldsymbol{z}^\mathsf{T} \left( \sum_{i=1}^n \lambda_i \boldsymbol{q}_i \boldsymbol{p}_i^\mathsf{T} \right) \boldsymbol{z} \right], \qquad (14)$$

$$= \sum_{i=1}^n \lambda_i \sum_{j=1}^s \boldsymbol{z}_j^\mathsf{T} \boldsymbol{q}_i \boldsymbol{p}_i^\mathsf{T} \boldsymbol{z}_j - \sum_{i=1}^n \lambda_i \mathbb{E}_{\boldsymbol{z}} \left[ \boldsymbol{z}^\mathsf{T} \boldsymbol{q}_i \boldsymbol{p}_i^\mathsf{T} \boldsymbol{z} \right], \qquad (15)$$

$$= \sum_{i=1}^n \lambda_i \left( \sum_{j=1}^s \boldsymbol{z}_j^\mathsf{T} \boldsymbol{q}_i \boldsymbol{p}_i^\mathsf{T} \boldsymbol{z}_j - \mathbb{E}_{\boldsymbol{z}} \left[ \boldsymbol{z}^\mathsf{T} \boldsymbol{q}_i \boldsymbol{p}_i^\mathsf{T} \boldsymbol{z} \right] \right), \qquad (16)$$

$$= \sum_{i=1}^n \lambda_i \left( \boldsymbol{q}_i^\mathsf{T} \left( \sum_{j=1}^s \boldsymbol{z}_j \boldsymbol{z}_j^\mathsf{T} \right) \boldsymbol{p}_i - \boldsymbol{q}_i^\mathsf{T} \underbrace{\mathbb{E}_{\boldsymbol{z}} \left[ \boldsymbol{z} \boldsymbol{z}^\mathsf{T} \right]}_{\mathbf{I}} \boldsymbol{p}_i \right), \qquad (17)$$

$$= \sum_{i=1}^n \lambda_i \boldsymbol{q}_i^\mathsf{T} \underbrace{\left( \left( \sum_{j=1}^s \boldsymbol{z}_j \boldsymbol{z}_j^\mathsf{T} \right) - \mathbf{I} \right)}_{\mathbf{M}} \boldsymbol{p}_i. \qquad (18)$$

Therefore, we can bound the norm of the difference as

$$|\tilde{g}_k(\theta) - g_k(\theta)| \leq \sum_{i=1}^{n} |\lambda_i| \left| \boldsymbol{q}_i^\mathsf{T} \mathbf{M} \boldsymbol{p}_i \right|, \tag{19}$$

$$\leq \sum_{i=1}^{n} |\lambda_i| \|\mathbf{M}\|_{\mathrm{op}} \tag{20}$$

where $\|\mathbf{M}\|_{\mathrm{op}}$ is the operator (spectral) norm of $\mathbf{M}$.

**Lemma 3** $\mathbb{P}\left[\|\mathbf{M}\|_{\mathrm{op}} > \beta\right] < \sum_{\boldsymbol{c} \in \Sigma_\epsilon} \frac{\mathbb{E}\left[\|\mathbf{M}\boldsymbol{c}\|^2\right]}{\beta^2(1-\epsilon)^2}$, *where $\Sigma_\epsilon$ is an $\epsilon$-net on an $\mathbb{R}^n$-sphere $\mathcal{S}^{n-1}$.*
**Proof** *We turn the lower bound $\|\mathbf{M}\|_{\mathrm{op}} > \beta$ on the spectral norm into a lower bound on $\mathbf{M}\boldsymbol{c}$ for any $\boldsymbol{c} \in \mathcal{S}^{n-1}$ which is a close approximation to the largest-norm eigenvalue eigenvector.*

*Consider the unit vector $\boldsymbol{u}^*$ such that $\|\mathbf{M}\boldsymbol{u}^*\| = \|\mathbf{M}\|_{\mathrm{op}}$. Such a vector exists $\|\mathbf{M}\|_{\mathrm{op}} = \sup_{\boldsymbol{u} \in \mathcal{S}^{n-1}} \boldsymbol{u}^\mathsf{T} \mathbf{M} \boldsymbol{u}$, because the supremum is taken over a compact subspace of $\mathbb{R}^n$. Note that if we have a unit vector $\boldsymbol{c} \in \mathbb{R}^n$ such that $\|\boldsymbol{c} - \boldsymbol{u}^*\| < \epsilon$, then*

$$\|\mathbf{M}\boldsymbol{c}\| \geq \|\mathbf{M}\boldsymbol{u}^*\| - \|\mathbf{M}(\boldsymbol{c} - \boldsymbol{u}^*)\|, \qquad \triangle \text{ triangle inequality} \tag{21}$$

$$= \|\mathbf{M}\|_{\mathrm{op}} - \|\mathbf{M}(\boldsymbol{c} - \boldsymbol{u}^*)\|, \tag{22}$$

$$\geq \|\mathbf{M}\|_{\mathrm{op}} - \|\mathbf{M}\|_{\mathrm{op}} \|\boldsymbol{c} - \boldsymbol{u}^*\|, \tag{23}$$

$$> \|\mathbf{M}\|_{\mathrm{op}} - \|\mathbf{M}\|_{\mathrm{op}} \epsilon = \|\mathbf{M}\|_{\mathrm{op}}(1 - \epsilon). \tag{24}$$

*Consider a finite $\epsilon$-net $\Sigma_\epsilon \subset \mathcal{S}^{n-1}$ on the unit sphere $\mathcal{S}^{n-1}$. There exists such a collection with cardinality at most $\left(1 + \frac{2}{\epsilon}\right)^n$, and thus such a finite $\epsilon$-net exists for $\epsilon > 0$. By (21), if $\|\mathbf{M}\|_{\mathrm{op}} > \beta$, then for some $\boldsymbol{c} \in \Sigma_\epsilon$ we have that $\|\mathbf{M}\boldsymbol{c}\| > \beta(1 - \epsilon)$. Hence*

$$\mathbb{P}\left[\|\mathbf{M}\|_{\mathrm{op}} > \beta\right] \leq \mathbb{P}\left[\cup_{\boldsymbol{c}_i \in \Sigma_\epsilon}\left[\|\mathbf{M}\boldsymbol{c}_i\| > \beta(1 - \epsilon)\right]\right], \tag{25}$$

$$\leq \sum_{\boldsymbol{c}_i \in \Sigma_\epsilon} \mathbb{P}\left[\|\mathbf{M}\boldsymbol{c}_i\| > \beta(1 - \epsilon)\right], \qquad \triangle \text{ union bound} \tag{26}$$

$$< \sum_{\boldsymbol{c}_i \in \Sigma_\epsilon} \frac{\mathbb{E}\left[\|\mathbf{M}\boldsymbol{c}_i\|^2\right]}{\beta^2(1 - \epsilon)^2}. \qquad \triangle \text{ Markov inequality} \tag{27}$$

$\blacksquare$

**Lemma 4** *For pairwise independent zero-mean identity-covariance $\boldsymbol{z}_j$ with pairwise independent coordinates, and any unit vector $\boldsymbol{c} \in \mathbb{R}^n$*

$$\mathbb{E}\left[\left\|\left(\frac{1}{s}\sum_p \boldsymbol{z}_p \boldsymbol{z}_p^\mathsf{T} - \mathbf{I}\right)\boldsymbol{c}\right\|^2\right] = \frac{\mathbb{E}[z^4] + n - 2}{s}. \tag{28}$$

**Proof**

$$\left\| \left( \frac{1}{s} \sum_p z_p z_p^\mathsf{T} - \mathbf{I} \right) c \right\|^2 \tag{29}$$

$$= \left\| \frac{1}{s} \sum_p z_p z_p^\mathsf{T} c - c \right\|^2 \tag{30}$$

$$= \left( \frac{1}{s} \sum_p z_p \left( z_p^\mathsf{T} c \right) - c \right)^\mathsf{T} \left( \frac{1}{s} \sum_q z_q \left( z_q^\mathsf{T} c \right) - c \right) \tag{31}$$

$$= \left( \frac{1}{s^2} \sum_{p,q} \left( z_p^\mathsf{T} c \right) z_p^\mathsf{T} z_q \left( z_q^\mathsf{T} c \right) \right) - \left( \frac{1}{s} \sum_p \left( z_p^\mathsf{T} c \right) \left( z_p^\mathsf{T} c \right) \right) - \left( \frac{1}{s} \sum_q \left( z_q^\mathsf{T} c \right) \left( z_q^\mathsf{T} c \right) \right) + c^\mathsf{T} c \tag{32}$$

$$= \left( \frac{1}{s^2} \sum_{p,q} \left( z_p^\mathsf{T} c \right) \left( z_q^\mathsf{T} c \right) z_p^\mathsf{T} z_q \right) - \left( \frac{2}{s} \sum_p \left( z_p^\mathsf{T} c \right)^2 \right) + c^\mathsf{T} c \tag{33}$$

$$= \left( \frac{1}{s^2} \sum_{p,q} \left( \sum_i z_{p,i} c_i \right) \left( \sum_j z_{q,j} c_j \right) \left( \sum_k z_{p,k} z_{q,k} \right) \right) - \left( \frac{2}{s} \sum_p \left( \sum_i z_{p,i} c_i \right)^2 \right) + c^\mathsf{T} c \tag{34}$$

$$= \left( \frac{1}{s^2} \sum_{p,q} \sum_{i,j,k} z_{p,i} z_{q,j} z_{p,k} z_{q,k} c_i c_j \right) - \left( \frac{2}{s} \sum_p \left( \sum_i z_{p,i} c_i \right) \left( \sum_j z_{p,j} c_j \right) \right) + 1 \tag{35}$$

$$= \left( \frac{1}{s^2} \sum_{p,q} \sum_{i,j,k} z_{p,i} z_{q,j} z_{p,k} z_{q,k} c_i c_j \right) - \left( \frac{2}{s} \sum_p \sum_{i,j} z_{p,j} z_{p,i} c_i c_j \right) + 1 \tag{36}$$

*Therefore, the expectation of the above can be simplified as*

$$\left( \frac{1}{s^2} \sum_{p,q,i,j,k} \mathbb{E} \left[ z_{p,i} z_{q,j} z_{p,k} z_{q,k} \right] c_i c_j \right) - \left( \frac{2}{s} \sum_{p,i,j} \underbrace{\mathbb{E} \left[ z_{p,j} z_{p,i} \right]}_{0 \ unless \ i \ = \ j} c_i c_j \right) + 1, \tag{37}$$

$$= \left( \frac{1}{s^2} \sum_{p,q,i,j,k} \mathbb{E} \left[ z_{p,i} z_{q,j} z_{p,k} z_{q,k} \right] c_i c_j \right) - \left( \frac{2}{s} \sum_{p,i} \overbrace{\mathbb{E} \left[ z_{p,i}^2 \right]}^{1} c_i^2 \right) + 1, \tag{38}$$

$$= \left( \frac{1}{s^2} \sum_{p,q,i,j,k} \underbrace{\mathbb{E} \left[ z_{p,i} z_{q,j} z_{p,k} z_{q,k} \right]}_{\substack{0 \ unless \ i \ = \ j, \ p \ = \ q \\ or \ i \ = \ j \ = \ k}} c_i c_j \right) - 2 + 1. \tag{39}$$

*The expectation in the terms of the first sum will be 0 if $i \neq j$: if $i \neq k$ then $\mathbb{E}\left[z_{p,i} z_{q,j} z_{p,k} z_{q,k}\right] = \cancel{\mathbb{E}\left[z_{p,i}\right]}^0 \mathbb{E}\left[z_{q,j} z_{p,k} z_{q,k}\right]$, and if $k = i \neq j$ then $\mathbb{E}\left[z_{p,i} z_{q,j} z_{p,k} z_{q,k}\right] = \cancel{\mathbb{E}\left[z_{q,j}\right]}^0 \mathbb{E}\left[z_{p,i} z_{p,k} z_{q,k}\right]$. Hence, for all non-zero terms $i = j$ and the first sum can be simplified as $\sum_{p,q,i,k} \mathbb{E}\left[z_{p,i} z_{q,i} z_{p,k} z_{q,k}\right]$.*

*Then, we again have four cases for the terms $\mathbb{E}\left[z_{p,i} z_{q,i} z_{p,k} z_{q,k}\right]$:*

$$\mathbb{E}\left[z_{p,i} z_{q,i} z_{p,k} z_{q,k}\right] = \begin{cases} \mathbb{E}[z_{p,i}^4] & i = k, q = p \\ \mathbb{E}[z_{p,i}^2]\mathbb{E}[z_{q,i}^2] = 1 & i = k, q \neq p \\ \mathbb{E}[z_{p,i}^2]\mathbb{E}[z_{p,k}^2] = 1 & i \neq k, q = p \\ \mathbb{E}[z_{p,i}]\mathbb{E}[z_{q,i}]\mathbb{E}[z_{p,k}]\mathbb{E}[z_{q,k}] = 0 & i \neq k, q \neq p \end{cases}, \tag{40}$$

*and so, separating the sum into these cases, we get*

$$\mathbb{E}\left[\left\|\frac{1}{s}\sum_p \boldsymbol{z}_p \boldsymbol{z}_p^\mathsf{T} \boldsymbol{c} - \boldsymbol{c}\right\|^2\right], \tag{41}$$

$$= \frac{1}{s^2}\left(\left(\sum_{p,i}\mathbb{E}\left[z_{p,i}^4\right] c_i^2\right) + \left(\sum_p \sum_{q \neq p}\sum_i c_i^2\right) + \left(\sum_p \sum_i \sum_{k \neq i} c_i^2\right)\right) - 1, \tag{42}$$

$$= \frac{1}{s^2}\left(\left(\mathbb{E}\left[z^4\right]\sum_{p,i} c_i^2\right) + s(s-1) + s(n-1)\right) - 1, \tag{43}$$

$$= \frac{s\mathbb{E}\left[z^4\right]}{s^2} + \frac{s^2 - s}{s^2} + \frac{s(n-1)}{s^2} - 1, \tag{44}$$

$$= \frac{\mathbb{E}[z^4] + s - 1 + n - 1 - s}{s} = \frac{\mathbb{E}[z^4] + n - 2}{s}. \tag{45}$$

$\blacksquare$

Combining the previous lemmas gives:

**Lemma 5** $\mathbb{P}\left[\|\mathbf{M}\|_{\text{op}} > \beta\right] < \left(1 + \frac{2}{\epsilon}\right)^n \frac{\mathbb{E}[z^4] + n - 2}{s\beta^2(1-\epsilon)^2}$ *for $\mathbf{M}$ as defined* (13), *for any $\beta, \epsilon > 0$, where $\mathbb{E}[z^4]$ is the fourth moment of the coordinates of $\boldsymbol{z}_j$.*
**Proof** *Combining Theorems 3 and 4, and noting that there exists an $\epsilon$-net of size at most $\left(1 + \frac{2}{\epsilon}\right)^n$ (Vershynin, 2012) yields the result.* $\blacksquare$

Finally, applying Theorem 5 to (19) gives the desired bound:

$$|\tilde{g}_k(\theta) - g_k(\theta)| \leq \sum_{i=1}^n |\lambda_i| \|\mathbf{M}\|_{\text{op}} < \left(1 + \frac{2}{\epsilon}\right)^n \frac{\mathbb{E}\left[z^4\right] + n - 2}{s\beta^2(1-\epsilon)^2}\left(\sum_{i=1}^n |\lambda_i|\right), \tag{46}$$

where $\left(\sum_{i=1}^n |\lambda_i|\right) < \lambda^{\max} n$. $\blacksquare$

Theorem 2 implies a bound on the norm of the gradient error on the optimisation domain $\Theta$ by a simple union bound over each coordinate of $\tilde{\boldsymbol{g}}(\boldsymbol{\theta})$.

**Theorem 6** *Under the assumptions of Theorem 2,*

$$\mathbb{P}\left[\|\tilde{\boldsymbol{g}}(\boldsymbol{\theta}) - \boldsymbol{g}(\boldsymbol{\theta})\| > \beta\right] < \delta \qquad if \qquad s > d_{\boldsymbol{\theta}}\left(1 + \frac{2}{\epsilon}\right)^n \frac{\mathbb{E}\left[z^4\right] + n - 2}{\delta\beta^2(1-\epsilon)^2}n\lambda^{\max}, \qquad (47)$$

*i.e. the approximate gradient $\boldsymbol{g}(\boldsymbol{\theta})$ will be within $\beta$ of the true gradient on the entire optimisation space $\Theta$ with probability at least $(1-\delta)$.*

Now, if $\tilde{\boldsymbol{g}}(\boldsymbol{\theta})$ is a conservative field, and so is implicitly a gradient of some (approximate) objective $\tilde{\mathcal{L}} : \Theta \to \mathbb{R}$, the above result allows us to bound the error on the solution found when optimising using the approximate gradient $\tilde{\boldsymbol{g}}$ instead of the actual gradient $\boldsymbol{g} = \nabla\mathcal{L}$. However, in general, $\tilde{\boldsymbol{g}}(\boldsymbol{\theta})$ need not be strictly conservative. In practice, since $\tilde{\boldsymbol{g}}(\boldsymbol{\theta})$ converges to a conservative field the more samples we take, we may assume that it is close enough to being conservative for the purposes of optimisation on hardware with finite numerical precision. Assuming that $\tilde{\boldsymbol{g}}(\boldsymbol{\theta})$ is conservative allows us to show the following bound on the optimum found when optimising using $\tilde{\boldsymbol{g}}(\boldsymbol{\theta})$, which is a restatement of Theorem 1:

**Theorem 7** *Let $\tilde{\boldsymbol{g}}$ and $\mathcal{L}$ be defined as in Theorem 2. Assume $\tilde{\boldsymbol{g}} : \Theta \to \mathbb{R}$ is a conservative field. Assume the optimisation domain $\Theta$ is convex, closed and bounded. Then, given a sufficiently large number of samples $s$, a maximum $\tilde{\boldsymbol{\theta}}^*$ obtained by maximising the objective implied by the approximate gradients $\tilde{\boldsymbol{g}}$ will be $\gamma$-close in terms of the true objective $\mathcal{L}$ to the true maximum $\boldsymbol{\theta}^*$ of the objective $\mathcal{L}$:*

$$\mathcal{L}(\tilde{\boldsymbol{\theta}}^*) \geq \mathcal{L}(\boldsymbol{\theta}^*) - \gamma \qquad if \qquad s > d_{\boldsymbol{\theta}}\left(1 + \frac{2}{\epsilon}\right)^n \frac{\mathbb{E}\left[z^4\right] + n - 2}{\delta\gamma^2(1-\epsilon)^2}n\lambda^{\max}\Delta\Theta, \qquad (48)$$

*with probability at least $1 - \delta$, where $\Delta\Theta \stackrel{\text{def}}{=} \sup_{\boldsymbol{\theta},\boldsymbol{\theta}'\in\Theta} \|\boldsymbol{\theta}' - \boldsymbol{\theta}\|$ is the maximum distance between two elements in $\Theta$.*

**Proof** *Let $\tilde{\mathcal{L}} : \Theta \to \mathbb{R}$ be an approximate objective implied by the gradient field $\tilde{\boldsymbol{g}}$, namely a scalar field such that $\nabla\tilde{\mathcal{L}} = \tilde{\boldsymbol{g}}$. Such a scalar field exists if $\tilde{\boldsymbol{g}}$ is a conservative field, and is unique up to a constant (which does not affect the optimum).*

*Assume that $s$ is sufficiently large such that the gradient difference $\|\tilde{\boldsymbol{g}} - \boldsymbol{g}\|$ is bounded by $\frac{\gamma}{\Delta\Theta}$ with probability at least $1 - \delta$. As per Theorem 6, this will be the case when*

$$s > d_{\boldsymbol{\theta}}\left(1 + \frac{2}{\epsilon}\right)^n \frac{\mathbb{E}\left[z^4\right] + n - 2}{\delta\gamma^2(1-\epsilon)^2}n\lambda^{\max}\Delta\Theta. \qquad (49)$$

*For any two points $\boldsymbol{\theta}, \boldsymbol{\theta}' \in \Theta$, with $\Delta\boldsymbol{\theta} \overset{\text{def}}{=} \boldsymbol{\theta}' - \boldsymbol{\theta}$, we have that*

$$\left| \left( \mathcal{L}(\boldsymbol{\theta}') - \mathcal{L}(\boldsymbol{\theta}) \right) + \left( \tilde{\mathcal{L}}(\boldsymbol{\theta}') - \tilde{\mathcal{L}}(\boldsymbol{\theta}) \right) \right| \tag{50}$$

$\triangle$ *Replace difference in values with integral along path from $\boldsymbol{\theta}$ to $\boldsymbol{\theta}'$*

$$= \left| \int_0^1 \frac{\partial}{\partial t} \mathcal{L} \left( \boldsymbol{\theta} + \Delta\boldsymbol{\theta}t \right) dt - \int_0^1 \frac{\partial}{\partial t} \tilde{\mathcal{L}} \left( \boldsymbol{\theta} + \Delta\boldsymbol{\theta}t \right) dt \right|, \tag{51}$$

$$= \left| \int_0^1 \Delta\boldsymbol{\theta} \cdot \nabla \mathcal{L} \left( \boldsymbol{\theta} + \Delta\boldsymbol{\theta}t \right) dt - \int_0^1 \Delta\boldsymbol{\theta} \cdot \nabla \tilde{\mathcal{L}} \left( \boldsymbol{\theta} + \Delta\boldsymbol{\theta}t \right) dt \right|, \tag{52}$$

$$= \left| \int_0^1 \Delta\boldsymbol{\theta} \cdot \left( \nabla \mathcal{L} - \nabla \tilde{\mathcal{L}} \right) \left( \boldsymbol{\theta} + \Delta\boldsymbol{\theta}t \right) dt \right|, \tag{53}$$

$$\leq \int_0^1 \left| \Delta\boldsymbol{\theta} \cdot \left( \nabla \mathcal{L} - \nabla \tilde{\mathcal{L}} \right) \left( \boldsymbol{\theta} + \Delta\boldsymbol{\theta}t \right) \right| dt, \tag{54}$$

$$= \int_0^1 \|\Delta\boldsymbol{\theta}\| \left\| \left( \nabla \mathcal{L} - \nabla \tilde{\mathcal{L}} \right) \left( \boldsymbol{\theta} + \Delta\boldsymbol{\theta}t \right) \right\| dt \leq \int_0^1 \|\Delta\boldsymbol{\theta}\| \left| \frac{\gamma}{\Delta\Theta} \right| dt \leq \gamma. \tag{55}$$

$\triangle$ *Difference of gradients bounded by Theorem 6*

$$\tag{56}$$

*Hence,*

$$\mathcal{L}(\boldsymbol{\theta}^*) - \mathcal{L}(\tilde{\boldsymbol{\theta}}^*) \leq \mathcal{L}(\boldsymbol{\theta}^*) - \mathcal{L}(\tilde{\boldsymbol{\theta}}^*) - \overbrace{\left( \tilde{\mathcal{L}}(\boldsymbol{\theta}^*) - \tilde{\mathcal{L}}(\tilde{\boldsymbol{\theta}}^*) \right)}^{\substack{\text{Negative because } \tilde{\boldsymbol{\theta}}^* \text{ is} \\ \text{a maximum of } \tilde{\mathcal{L}}}}, \tag{57}$$

$$\leq \left| \mathcal{L}(\boldsymbol{\theta}^*) - \mathcal{L}(\tilde{\boldsymbol{\theta}}^*) - \left( \tilde{\mathcal{L}}(\boldsymbol{\theta}^*) - \tilde{\mathcal{L}}(\tilde{\boldsymbol{\theta}}^*) \right) \right| \leq \gamma, \tag{58}$$

*which gives the bound in the theorem.* ∎

## Appendix B. Implementation Details

Our implementation uses the `JAX` library (Bradbury et al., 2018) and all experiments were conducted on A100 GPUs using double floating point precision. The `softplus` function was used to enforce positive value constraints during hyperparameter optimisation. During each step of marginal likelihood optimisation, the linear system solvers were run until all linear systems in the batch reached a relative residual norm $\|\mathbf{H}_{\boldsymbol{\theta}}\boldsymbol{v} - \boldsymbol{b}\|/\|\boldsymbol{b}\|$ of less than $\epsilon_{\text{rel}}^{\text{mean}} = 0.01$ for the linear system $\mathbf{H}_{\boldsymbol{\theta}}\boldsymbol{v_y} = \boldsymbol{y}$, corresponding to the mean, and $\epsilon_{\text{rel}}^{\text{samples}} = 0.1$ for the linear systems $\mathbf{H}_{\boldsymbol{\theta}}\boldsymbol{v_j} = \boldsymbol{z_j}$, corresponding to the samples. Conjugate gradients and alternating projections keep track of the residual as part of the algorithm. For stochastic gradient descent, we estimate the current residual by keeping a residual vector in memory and updating it sparsely whenever we compute the gradient on a mini-batch of data, leveraging the property that the gradient is equal to the residual. In practice, we find that this estimates an approximate upper bound on the true residual. For conjugate gradients, we did not use any preconditioner. For alternating projections, we used a block size of 2000. For stochastic gradient descent, we used a mini-batch size of 1000, momentum of 0.9, no Polyak averaging, and learning rates of 90, 20, 100, 20, and 30 respectively for POL, ELEVATORS, BIKE, PROTEIN and KEGGDIRECTED, which were selected by performing a grid search.

## Appendix C. Additional Empirical Results

Table 2: Test root-mean-square errors, test log-likelihoods, total runtimes and solver runtimes in minutes after 100 steps of marginal likelihood optimisation, and speed-up per dataset due to warm start (mean $\pm$ standard error over 10 dataset splits).

| | | Test RMSE | Test LLH | Total Runtime | Solver Runtime | Speed-Up |
|---|---|---|---|---|---|---|
| **POL** $n = 13500, d = 26$ | CG | $0.075 \pm 0.001$ | $1.268 \pm 0.008$ | $7.857 \pm 0.111$ | $7.641 \pm 0.110$ | — |
| | + ws | $0.075 \pm 0.001$ | $1.268 \pm 0.009$ | $2.003 \pm 0.027$ | $1.790 \pm 0.026$ | $3.9 \times$ |
| | AP | $0.075 \pm 0.001$ | $1.269 \pm 0.008$ | $22.390 \pm 0.331$ | $22.158 \pm 0.326$ | — |
| | + ws | $0.075 \pm 0.001$ | $1.268 \pm 0.009$ | $0.993 \pm 0.015$ | $0.780 \pm 0.014$ | $22.6 \times$ |
| | SGD | $0.075 \pm 0.001$ | $1.266 \pm 0.010$ | $41.306 \pm 0.201$ | $41.215 \pm 0.201$ | — |
| | + ws | $0.075 \pm 0.001$ | $1.268 \pm 0.007$ | $3.077 \pm 0.016$ | $2.989 \pm 0.016$ | $13.4 \times$ |
| **ELEVATORS** $n = 14940, d = 18$ | CG | $0.355 \pm 0.003$ | $-0.386 \pm 0.007$ | $2.758 \pm 0.044$ | $2.542 \pm 0.042$ | — |
| | + ws | $0.355 \pm 0.003$ | $-0.386 \pm 0.007$ | $1.072 \pm 0.014$ | $0.858 \pm 0.012$ | $2.6 \times$ |
| | AP | $0.355 \pm 0.003$ | $-0.386 \pm 0.007$ | $13.547 \pm 0.345$ | $13.331 \pm 0.344$ | — |
| | + ws | $0.355 \pm 0.003$ | $-0.386 \pm 0.007$ | $0.516 \pm 0.006$ | $0.303 \pm 0.004$ | $26.2 \times$ |
| | SGD | $0.355 \pm 0.003$ | $-0.385 \pm 0.007$ | $4.921 \pm 0.069$ | $4.685 \pm 0.015$ | — |
| | + ws | $0.355 \pm 0.003$ | $-0.386 \pm 0.007$ | $0.980 \pm 0.065$ | $0.748 \pm 0.004$ | $5.2 \times$ |
| **BIKE** $n = 15642, d = 17$ | CG | $0.033 \pm 0.003$ | $2.150 \pm 0.018$ | $7.689 \pm 0.128$ | $7.451 \pm 0.126$ | — |
| | + ws | $0.033 \pm 0.003$ | $2.150 \pm 0.017$ | $2.180 \pm 0.038$ | $1.945 \pm 0.036$ | $3.5 \times$ |
| | AP | $0.033 \pm 0.003$ | $2.151 \pm 0.018$ | $12.306 \pm 0.210$ | $12.068 \pm 0.207$ | — |
| | + ws | $0.033 \pm 0.003$ | $2.153 \pm 0.018$ | $0.904 \pm 0.014$ | $0.670 \pm 0.012$ | $13.6 \times$ |
| | SGD | $0.033 \pm 0.003$ | $2.179 \pm 0.020$ | $81.843 \pm 1.373$ | $81.676 \pm 1.372$ | — |
| | + ws | $0.032 \pm 0.003$ | $2.149 \pm 0.031$ | $6.733 \pm 0.168$ | $6.567 \pm 0.168$ | $12.2 \times$ |
| **PROTEIN** $n = 41157, d = 9$ | CG | $0.503 \pm 0.004$ | $-0.587 \pm 0.010$ | $31.438 \pm 0.476$ | $29.850 \pm 0.458$ | — |
| | + ws | $0.503 \pm 0.004$ | $-0.588 \pm 0.010$ | $11.270 \pm 0.156$ | $9.685 \pm 0.138$ | $2.8 \times$ |
| | AP | $0.503 \pm 0.004$ | $-0.587 \pm 0.010$ | $45.417 \pm 0.622$ | $43.829 \pm 0.607$ | — |
| | + ws | $0.503 \pm 0.004$ | $-0.587 \pm 0.010$ | $5.519 \pm 0.068$ | $3.934 \pm 0.053$ | $8.2 \times$ |
| | SGD | $0.504 \pm 0.004$ | $-0.587 \pm 0.010$ | $46.915 \pm 0.350$ | $44.661 \pm 0.349$ | — |
| | + ws | $0.504 \pm 0.004$ | $-0.589 \pm 0.009$ | $7.874 \pm 0.024$ | $5.621 \pm 0.024$ | $6.0 \times$ |
| **KEGGDIRECTED** $n = 43945, d = 20$ | CG | $0.084 \pm 0.002$ | $1.082 \pm 0.017$ | $64.290 \pm 0.768$ | $61.902 \pm 0.760$ | — |
| | + ws | $0.084 \pm 0.002$ | $1.081 \pm 0.017$ | $18.807 \pm 0.228$ | $16.415 \pm 0.220$ | $3.4 \times$ |
| | AP | $0.084 \pm 0.002$ | $1.082 \pm 0.017$ | $62.235 \pm 0.625$ | $59.848 \pm 0.618$ | — |
| | + ws | $0.084 \pm 0.002$ | $1.081 \pm 0.018$ | $4.857 \pm 0.054$ | $2.464 \pm 0.046$ | $12.8 \times$ |
| | SGD | $0.084 \pm 0.002$ | $1.081 \pm 0.019$ | $360.436 \pm 4.079$ | $357.734 \pm 4.076$ | — |
| | + ws | $0.084 \pm 0.002$ | $1.073 \pm 0.014$ | $48.721 \pm 0.548$ | $46.020 \pm 0.545$ | $7.4 \times$ |

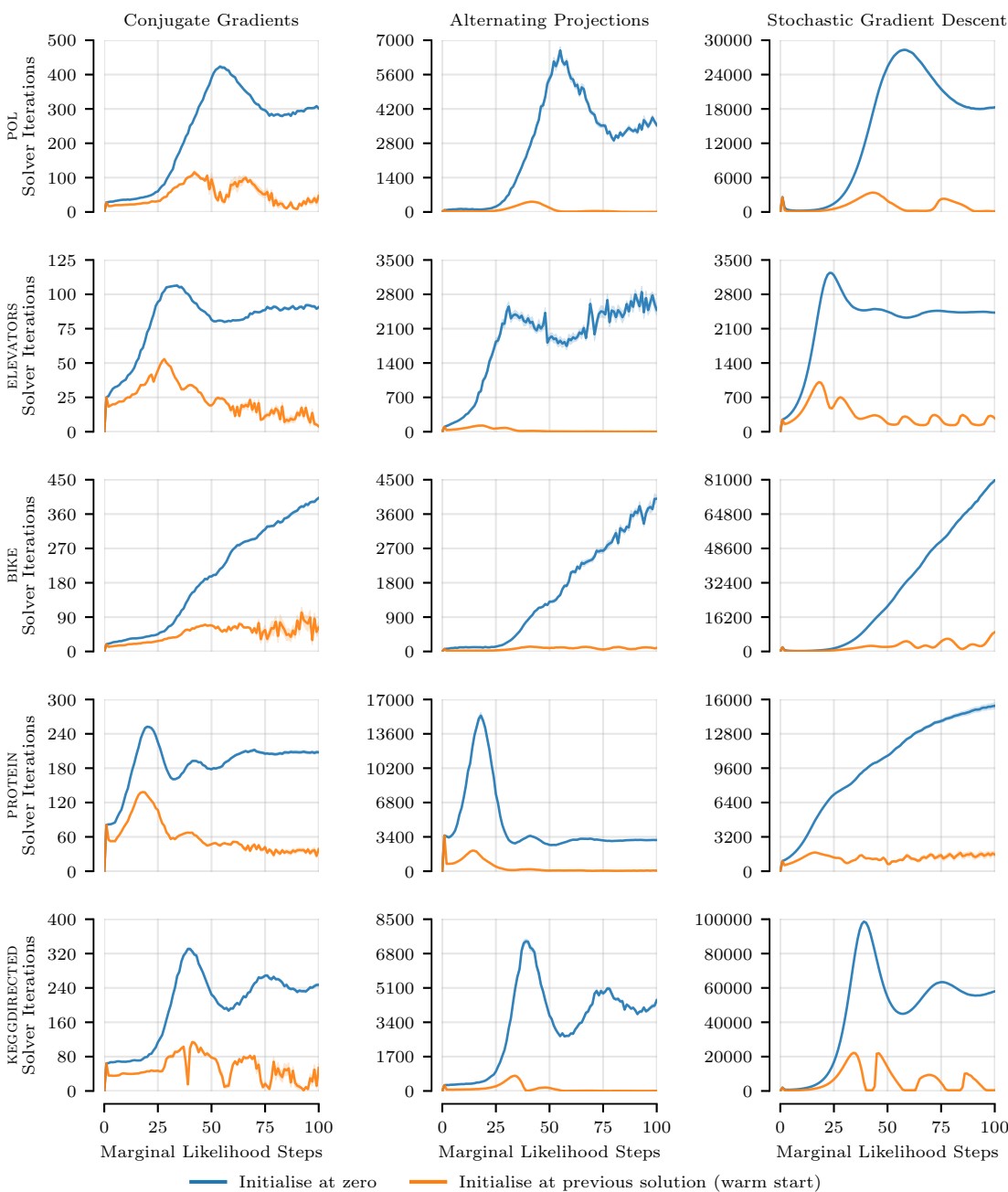

Figure 6: Evolution of the required number of linear system solver iterations at each step of marginal likelihood optimisation on different datasets. Initialising at the solution of the previous step reduces the number of required solver iterations with varying effectiveness among different solvers and datasets.

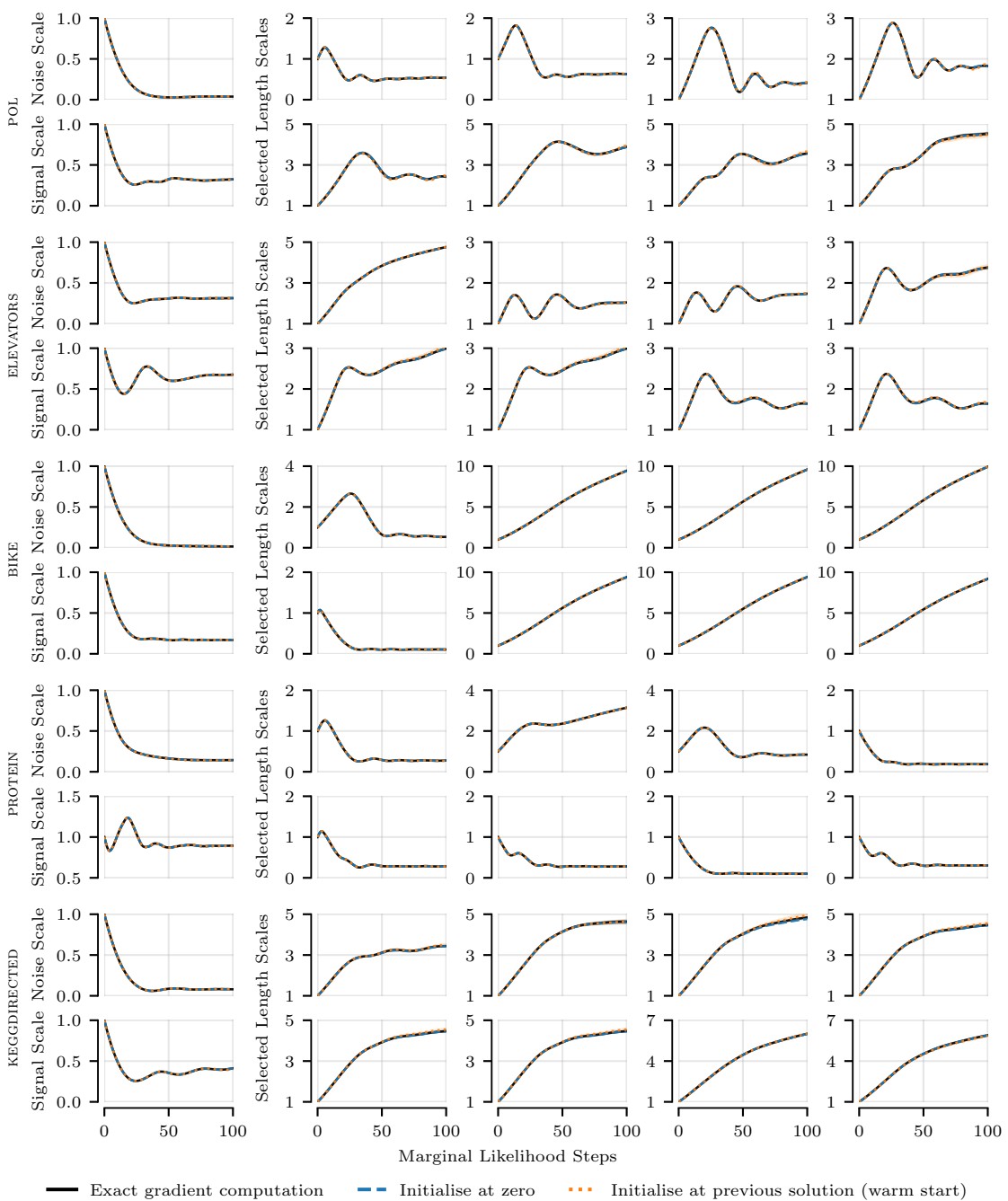

Figure 7: Evolution of hyperparameters during marginal likelihood optimisation on different datasets using conjugate gradients as linear system solver. The behaviour of exact gradient computation using Cholesky factorisation is obtained when initialising at zero or at the previous solution (warm start). The latter does not degrade performance.

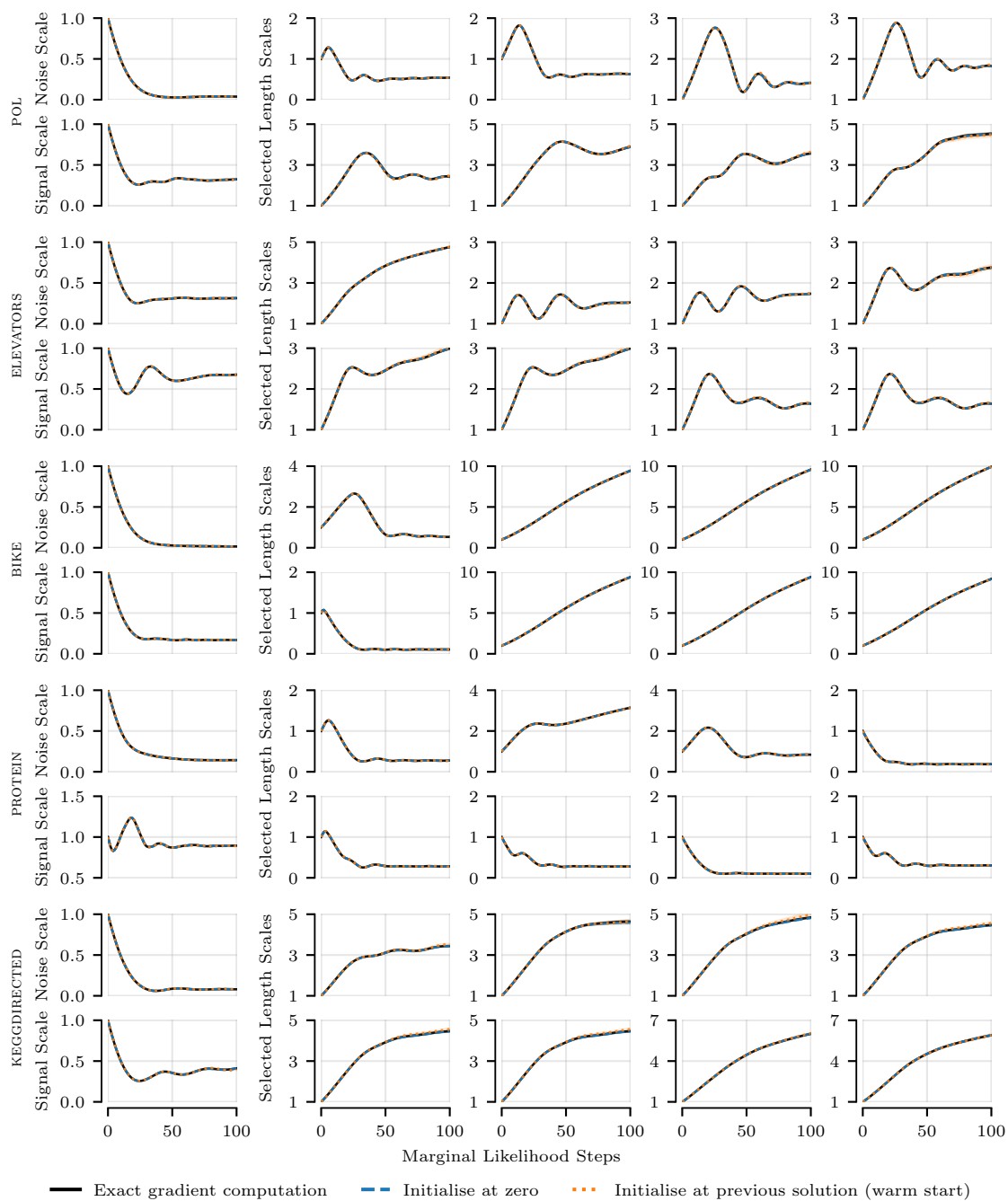

Figure 8: Evolution of hyperparameters during marginal likelihood optimisation on different datasets using alternating projections as linear system solver. The behaviour of exact gradient computation using Cholesky factorisation is obtained when initialising at zero or at the previous solution (warm start). The latter does not degrade performance.

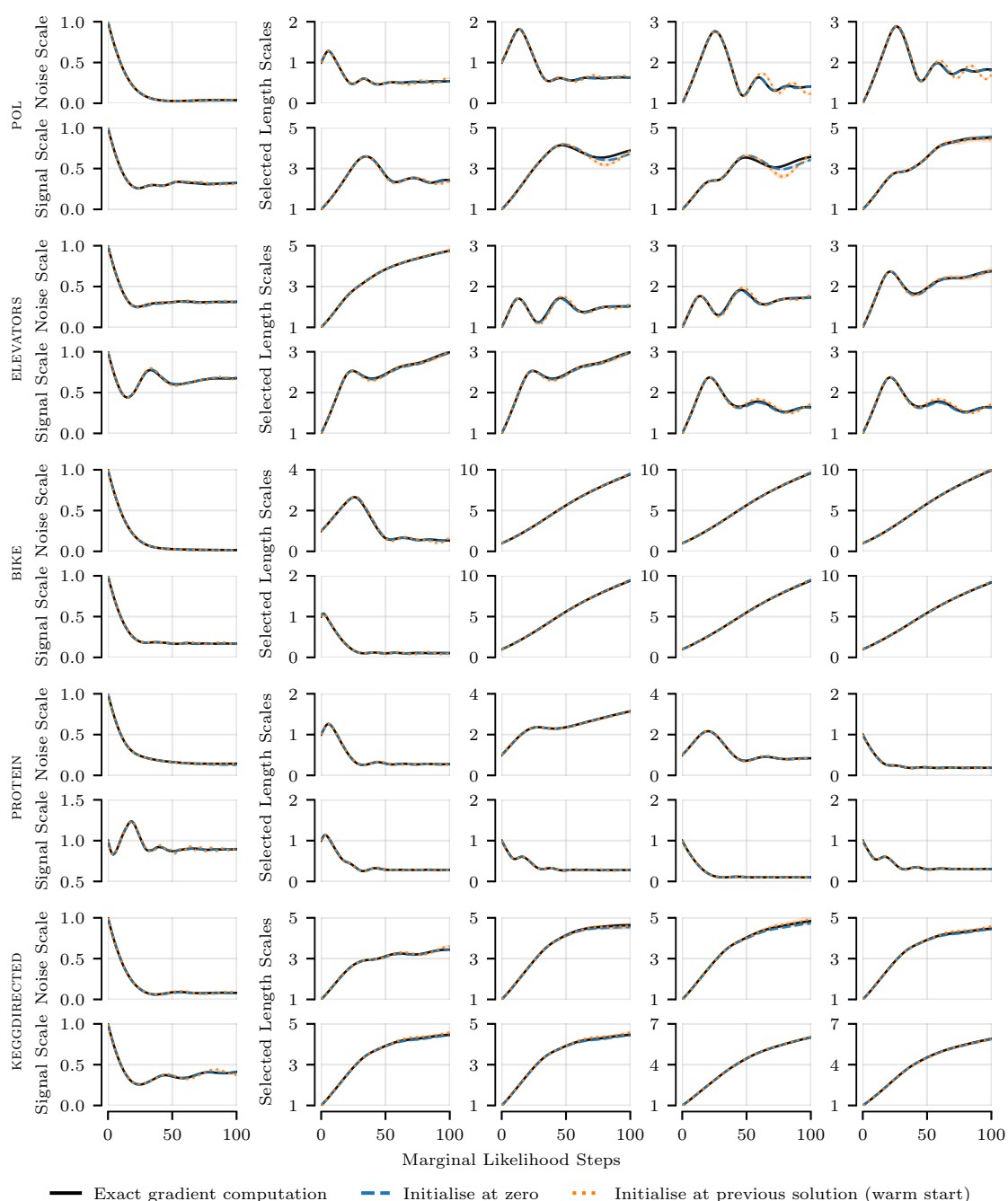

Figure 9: Evolution of hyperparameters during marginal likelihood optimisation on different datasets using stochastic gradient descent as linear system solver. The behaviour of exact gradient computation using Cholesky factorisation is obtained when initialising at zero or at the previous solution (warm start). The latter does not degrade performance.

