# OpenReview forum: "Warm Start Marginal Likelihood Optimisation for Iterative Gaussian Processes"
_approximateinference.org/AABI/2024/Symposium — AABI 2024_

### Official Review · Reviewer_ysiL · 2024-04-19
**Improving Convergence Time of Iterative Gaussian Processes using Warm Starts**

**Rating:** 8
**Confidence:** 4

**Review:**

The work is well written and clear on its contribution to the field. The paper demonstrated that the linear system solver to find the inverse of the kernel matrix (N x N) times a vector is the bottleneck in computation time. They cleverly use a Taylor expansion of the gradient update on the kernel matrix in order to reuse solutions from the linear system solved in the previous gradient step. They show this has a large impact in computation time and number of iterations. At the same time, they demonstrate empirically that this has virtually no downside with the solution found, and the whole optimization trajectory, essentially the same as without using the warm start strategy. This I found quite surprising. The only thing I may have wanted to see if a plot of how the number of probe vectors affect the learning with the warm start. It seems fixing the probe vectors in the warm start method might make the method sensitive to the number of probe vectors used. Overall, this was a nice read and good work.

---

### Official Review · Reviewer_8mLC · 2024-04-21
**Impressive results**

**Rating:** 7
**Confidence:** 3

**Review:**

**- Summary**
The authors organize marginal likelihood optimization for iterative Gaussian processes into a three-level hierarchy and propose using a Taylor expansion to obtain a warm start for the next iteration from the last iteration.

**- Pros**
- The improvement in scalability is impressive on real data, with detailed theorems provided on the bound (seemingly correct upon a very rough check).

**- Questions**
I don't see any cons of this work, but I do have some open questions:
- The baseline comparison uses an initialization at zero. I am not sure whether this is common practice, as the zero value initial point seems somewhat naive.
- I am unsure whether the property of $\boldsymbol H$ holds for all types of kernels $k(\cdot, \cdot)$.


**- Conclusion**
Overall, I believe the paper presents impressive improvement in scalability for Gaussian processes. While the questions I have raised may not be essential for the validation of the results, addressing them could provide additional insights and potentially broaden the applicability of the findings.

---

### Official Review · Reviewer_m9QG · 2024-04-24
**A new efficient method for Marginal Likelihood Optimisation for Iterative Gaussian Processes.**

**Rating:** 7
**Confidence:** 3

**Review:**

The authors provide a new method for the marginal likelihood optimization problem
for iterative Gaussian processes. The manuscript provides sufficient background and describes the relevant previous works.

The main idea is not new in the field of optimization but as far as I understand it is for first time applied in this context. It boils down in reusing solutions of linear systems in the course of the marginal likelihood optimization (where linear system solver dominates the overall computational costs).

Interestingly, the authors prove the correctness of their method and also provide experimental evidence for the speed up that their method offers over conventional methods. However, it would be useful to make their code available for the benefit of the community as well as to strengthen their results (reproducibility).

Overall, the manuscript seems interesting, novel and impactful.

---

### Official Review · Reviewer_KUVz · 2024-04-25

**Rating:** 7
**Confidence:** 3

**Review:**

In this paper the authors propose a method for speeding up the optimization of Gaussian Process hyperparameters.  In particular, the standard approach to optimization requires solving linear systems at each iteration, but these linear systems do not change much from iteration to iteration.  As such, the authors propose using iterative solvers initialized at the solution found in the previous iteration of the hyperparameter optimization as a warm start.  This generally results in a meaningful speed up (~3-10x), but comes at a slight cost in accuracy because in order to solve (nearly) the same linear system at each iteration, the same probe vectors must be used across all iterations in the randomized trace estimator, as opposed to drawing new probe vectors at each iteration.  The amount of error (in terms of the objective function) that this introduces is theoretically bounded by the authors.

Overall, I found the paper to be well-explained, clear, and interesting.  The idea of using the warm start feels a bit simple to me, but there are sufficient complications arising from the randomized trace estimation that I found the results non-trivial.  I do have some comments below in the hopes that they're helpful to the authors:

- I'm wondering if the theoretical bias arising from reusing the probe vectors can be ameliorated in some simple way.  For instance, one could imagine that every $k$ iterations, new probe vectors are drawn.  This obviously would require solving the linear system from scratch ever $k$ iterations, but warm starts could still be used for all other iterations.  Another possibility would be to slightly perturb the probe vectors at each iteration in such a way that from iteration to iteration the linear systems remain close enough to enjoy the advantage of the warm start, but over many iterations the probe vectors become essentially independent of their starting value.  One example would be to take Gaussian probe vectors and at each iteration obtain new probe vectors via $z_{j}^{(t)} = \sqrt{(1-\epsilon)} z_j^{(t-1)} + \sqrt{\epsilon} \psi_{j}^{(t)} $ for some small $\epsilon$ with $\psi_j ^{(t)} \sim \mathcal{N}(0, 1)$ drawn anew at each iteration.  This scheme would result in the entries of the probe vectors always being marginally distributed $\mathcal{N}(0, 1)$, but without changing too much from iteration to iteration.
- Theorem 1 is a nice result, but in terms of prediction isn't it more important to show that the inferred hyperparameters are close, as opposed to the objective functions being close?
- Presumably Theorem 2 is incredibly loose in terms of its sample complexity.  If I understand the result correctly, one would need a number of probe vectors exponential in the sample size, which is far more than the number used in the numerical results (which seems to produce highly accurate results).
- Perhaps relatedly, and I may have missed this, but what is $\epsilon$ in Theorem 2?  The statement of the theorem says that it holds for any $\epsilon > 0$, but that cannot be true (e.g., taking $\epsilon \rightarrow \infty$ would result in the requirement $s > 0$.  I believe (from the proof of Lemma 3) that $0 < \epsilon < 1$.
- The proof of Theorem 2 refers to a "Theorem 5" which I believe should be "Lemma 5"

---

### Official Review · Reviewer_Q8Mw · 2024-04-25
**Warm Start....Review**

**Rating:** 7
**Confidence:** 3

**Review:**

Overview:

This paper proposes a new algorithm for optimizing the marginal likelihood for iterative Gaussian processes. The main contribution is a reformulation of the gradient term that allows for a warm start approach to solving equations at each optimization step. This technique is applied to conjugate gradient, alternating projects, and stochastic gradient descent methods on several datasets, displaying a promising speed-up over traditional methods.

Overall, the paper is well-written and has a good progression of ideas in an engaging and informative way.

Main Comments:

The main contribution of the paper is a warm start method for solving a linear system which is derived from the optimization of the marginal likelihood of a gaussian process. Although this idea is novel in this context, warm-starting in Bayesian optimization is a well-studied problem. I would appreciate it if the authors could draw connections to similar methods.

In theorem 1, the authors assume that the hyperparameter optimization domain \Theta is convex, closed, and bounded. These assumptions are very strong, and it would be nice to have more justification for why they are acceptable assumptions or pose a weaker version of the theorem in the non-convex case.

A key element of the approach is that the change in (H_\theta)^{-1} changes very little between the timesteps; this is justified by the Taylor expansions in equations 10 and 11, but it seems this could be better characterized.

Minor Comments:

Theorem 2 in the appendix was very important for my understanding of the results, and it would be nice if it could be included in the main body. This could be either as a second main body theorem or incorporated into the statement of Theorem 1.

Another sentence or two at the end of section 2 explaining why, for large n Cholesky factorization, it is not tractable would help motivate the author’s approach.

The authors use “n” as the dimension of the problem in the main body and “n” as the number of samples in the appendix. This is stated in the text, but I think using “N” for one of the quantities would add to the paper's readability.

---

### Official Review · Reviewer_KZXe · 2024-04-25

**Rating:** 7
**Confidence:** 4

**Review:**

The paper discusses a method to improve convergence speed in the optimization of the Gaussian process (GP) marginal likelihood using linear systems solvers.
The method consists in performing warm-starts of the optimization by using the solution of the previous iteration.


- The paper is well written and easy to follow.
- The method is simple and intuitively it makes sense. The theoretical analysis is also clear and well done.
- The experiments are well designed and the results are convincing.
- The only small downside of the paper is not discussing preconditioning [e.g. 1], which is a common technique to improve convergence speed in CG. It would be interesting to see how the method compares to various preconditioning techniques and what happens when the method is combined with them.
- Why no abstract? I don't think I've seen relaxations from the organizers about this.


[1] Cutajar et al. Preconditioning Kernel Matrices

---

### Meta-Review · Area_Chair_GABY · 2024-05-13

**Recommendation:** Accept (Poster)
**Confidence:** 5

**Metareview:**

The paper develops a new method to speed up the marginal likelihood optimization in iterative Gaussian processes. The key idea is to reuse the intermediate solutions as warm starts. All reviewers found the paper interesting.

---

### Decision · Program_Chairs · 2024-05-27

Accept